# Functional Progression in Patients with Interstitial Lung Disease Resulted Positive to Antisynthetase Antibodies: A Multicenter, Retrospective Analysis

**DOI:** 10.3390/jcm9093033

**Published:** 2020-09-21

**Authors:** Giulia Dei, Paola Rebora, Martina Catalano, Marco Sebastiani, Paola Faverio, Maria Rosa Pozzi, Andreina Manfredi, Paolo Cameli, Francesco Salton, Carlo Salvarani, Lorenzo Cavagna, Marco Confalonieri, Elena Bargagli, Fabrizio Luppi, Alberto Pesci

**Affiliations:** 1Department of Medicine and Surgery, University of Milan-Bicocca, 20126 Milan, Italy; giulia.dei@hotmail.it (G.D.); m.catalano17@campus.unimib.it (M.C.); paola.faverio@unimib.it (P.F.); fabrizio.luppi@unimib.it (F.L.); 2Respiratory Unit, San Gerardo Hospital, 20900 Monza, Italy; 3Center of Biostatistics for Clinical Epidemiology, School of Medicine and Surgery, University of Milano-Bicocca, 20126 Milan, Italy; paola.rebora@unimib.it; 4Rheumatology Unit, University of Modena and Reggio Emilia, Azienda Ospedaliero-Universitaria Policlinico di Modena, 41125 Modena, Italy; marco.sebastiani@unimore.it (M.S.); Andreina.Manfredi@gmail.com (A.M.); carlo.salvarani@unimore.it (C.S.); 5Rheumatology Unit, S. Gerardo Hospital, 20900 Monza, Italy; m.pozzi@hsgerardo.org; 6Respiratory Disease and Lung Transplant Unit, Department of Medical Sciences, Surgery and Neurosciences, University of Siena, 53100 Siena, Italy; paolocameli88@gmail.com (P.C.); bargagli2@gmail.com (E.B.); 7Pneumology Unit, Department of Medical, Surgical and Health Sciences, University of Trieste, 34127 Trieste, Italy; francesco.salton@gmail.com (F.S.); marco.confalonieri@asugi.sanita.fvg.it (M.C.); 8Department of Rheumatology, University and IRCCS Policlinico S. Matteo Foundation of Pavia and ERN ReCONNET, 27100 Pavia, Italy; cavagna@unipv.it

**Keywords:** interstitial lung disease, antisynthetase syndrome, lung function, autoantibodies antisynthetase, anti-Jo-1 antibodies, no anti-Jo-1 antibodies

## Abstract

Antisynthetase syndrome (ASSD) is a rare autoimmune disease characterized by serologic positivity for antisynthetase antibodies. Anti-Jo1 is the most frequent, followed by anti PL-7, anti PL-12, anti EJ, and anti OJ antibodies. The lung is the most frequently affected organ, usually manifesting with an interstitial lung disease (ILD), which is considered the main determinant of prognosis. Some evidences suggest that non-anti-Jo-1 antibodies may be associated with more severe lung involvement and possibly with poorer outcomes, while other authors do not highlight differences between anti-Jo1 and other antisynthetase antibodies. In a multicenter, retrospective, “real life” study, we compared lung function tests (LFTs) progression in patients with ILD associated with anti-Jo1 and non-anti-Jo1 anti-synthetase antibodies to assess differences in lung function decline between these two groups. Therefore, we analyzed a population of 57 patients (56% anti-Jo1 positive), referred to the outpatient Clinic of four referral Centers in Italy (Modena, Monza, Siena, and Trieste) from 2008 to 2019, with a median follow-up of 36 months. At diagnosis, patients showed a mild ventilatory impairment and experienced an improvement of respiratory function during treatment. We did not observe statistically significant differences in LFTs at baseline or during follow-up between the two groups. Moreover, there were no differences in demographic data, respiratory symptoms onset (acute vs. chronic), extrapulmonary involvement, treatment (steroid and/or another immunosuppressant), or oxygen supplementation. Our study highlights the absence of differences in pulmonary functional progression between patients positive to anti-Jo-1 vs. non anti-Jo-1 antibodies, suggesting that the type of autoantibody detected in the framework of ASSD does not affect lung function decline.

## 1. Introduction

Antisynthetase syndrome (ASSD) is a rare autoimmune disease of unknown etiology, characterized by serologic positivity to antisynthetase antibodies [1] and typically involving lungs, joints, and muscles [2].

To date, at least 10 antisynthetase antibodies have been identified. The most frequent are the anti-Jo1, followed by the anti-EJ, anti-PL7, anti-PL12, and anti-OJ, while anti-KS, anti-Zo, anti-Yrs, anti-SC, and anti-JS are less frequently detected [3]. Anti-Ro 52 kDa antibodies are detected in about 30–50% of patients with ASSD, independently of the underlying antisynthetase antibody specificity (2) and in some previous reports, they have been related to a more severe interstitial lung disease (ILD), with increased prevalence of acute respiratory failure and development of lung fibrosis [4,5].

The lung is the most frequently affected organ, with development of diffuse ILD, associated with a ventilatory restrictive pattern and a reduction in carbon monoxide diffusion capacity (DLCO), also in the absence of other specific features, such as myositis, arthritis, mechanic’s hands, fever, and Raynaud’s phenomenon [6]. ILD represents the most severe organ involvement, leading to a high morbidity and mortality [7].

In some studies, patients with ASSD experienced a significant decline in ventilatory function as compared with healthy controls [6,8]. In contrast, in other studies, only a minority of patients progressed to advanced lung disease and most patients had stable pulmonary involvement for a long period of time [9,10].

Anti Jo-1 antibodies are considered the main markers of the ASSD [11]. However, according to previous studies, antisynthetase antibodies other than anti-Jo1 would be associated with a worse functional pattern and survival [12,13]. In contrast, some other studies did not observe differences in prognosis in relation to different antibody patterns (2) [14]. Despite conflicting results, it is possible that antisynthetase antibodies specificity may influence the functional outcome of these patients.

The aim of our study is to compare, in a multicenter, retrospective, and “real life” study, the lung function tests progression in patients with ILD associated with anti-Jo1 and non-anti-Jo1 antisynthetase antibodies and to assess differences in prognosis between the two groups.

## 2. Methods

### 2.1. Study Design

In this multicentre study, we retrospectively reviewed the clinical records of patients that received a diagnosis of ILD confirmed with high resolution computed tomography (HRCT) in the framework of an ASSD from 2008 to 2018. In the study, we included all patients with an ILD confirmed by HRCT and with a double confirmed positivity for antisynthetase antibodies, with at least one obtained in the laboratory of four Italian hospitals (Modena, Monza, Siena, Trieste) [15]. Demographic, clinical, serological, functional, and therapeutic data were collected. The follow-up was performed through a serial evaluation of lung function tests that have been performed, on average, every four months for the first two years. The latest available follow-up dates back to December 2019.

Disease progression and treatment response on pulmonary function tests (PFTs) were defined as a decrease or increase in FVC by more or less than 10% compared to the baseline, respectively, and/or a decrease or increase in DLCO by more or less than 15% of those predicted, respectively, similar to the established criteria for ILD in systemic sclerosis [16]. Cause of death was collected and registered. 

An acute exacerbation was defined according to Collard and colleagues as an acute, clinically significant deterioration of an unidentifiable cause in a patient with underlying IPF [17]. The local IRB approved the study.

### 2.2. Pulmonary Function Tests

The following lung function measurements were recorded according to ATS/ERS standards [18,19] using a plethysmograph with corrections for temperature and barometric pressure: forced expiratory volume in the first second (FEV1), forced vital capacity (FVC), FEV1/FVC ratio, total lung capacity (TLC), and carbon monoxide lung transfer factor (TLCO). A six-minute walking test was performed according to international recommendations [20]. These measurements were collected in all patients that were able to properly perform lung function tests, excluding patients with an acute lung disease.

Outcome data regarding clinical outcome were retrieved from regional databases.

### 2.3. Statistical Analysis

Continuous variables are reported as median and interquartile ranges (IQR), while categorical variables as numbers and percentages. To assess the differences between patients with anti-Jo1 and non-anti-Jo1 antisynthetase antibodies at baseline, we used the Fisher exact test for the categorical variables and the Mann-Whitney test for the continuous variables.

Changes over time in pulmonary function tests were analyzed with longitudinal mixed models including, as a response variable, the pulmonary function tests from baseline to follow-up. The dependence among repeated observation on the same subject was accounted for by the inclusion of a random intercept in the models. In terms of regressors, the models included the time of visit as a continuous variable (expressed as months from diagnosis and modeled by the use of polynomials when needed), the group (anti-Jo1), and, if significant, the interaction between the group and time. According to possible clinical implications, the models were adjusted for the pre-specified variables patient age, anti-SSA Ro 52 positivity, acute (ARDS—like) onset, diagnostic delay of more than 1 year, and immunosuppressive treatment.

## 3. Results

### 3.1. Differences between Anti-Jo1-Positive and Negative Subgroups at Baseline

A population of 57 patients with ASSD characterized by pulmonary involvement was recruited.

Thirty-two (56%) patients showed an anti-Jo1 positivity, 14 (25%) were anti PL-12 positive, six (11%) anti-PL7 positive, three (5%) anti-EJ positive, and two (4%) were anti-OJ positive.

Demographic, clinical, serological features, and lung function parameters at baseline are reported in Table 1.

There were no statistically significant differences in sex, smoking status, coexistence of other autoimmune diseases, and age at the time of ASSD diagnosis between anti-Jo1 and non-anti-Jo1 positive patients. No differences were also observed for the presence of other autoantibodies in the two groups, except for anti-SSA/Ro52+ positivity, which was more frequently reported in non-anti-Jo1 patients (76% versus 40% *p* = 0.019).

At baseline, no statistically significant differences in lung function tests, specifically in FVC, FEV1, TLC, and DLCO, were observed between the two groups. 

No significant differences were recorded in the clinical presentation of respiratory symptoms (cough, dyspnea, at rest, or on exertion) between the two groups. Moreover, there were no differences in respiratory symptoms onset (acute vs. chronic).

No statistically significant differences were observed regarding the frequency of extrapulmonary involvement (articular, muscular, gastrointestinal, cutaneous symptoms, and pulmonary hypertension) except for Raynaud’s phenomenon, which was more frequently observed in non-anti-Jo1 patients (29.2 versus 6.2, *p* = 0.03).

Finally, there were no differences of treatment (steroid and/or another immunosuppressant) or oxygen supplementation at baseline between the two groups.

### 3.2. Differences between Anti-Jo1 Positive and Negative Subgroups During Follow-Up

During the follow-up, three patients died (1 non anti-Jo1 and 2 anti-Jo1 positive, *p*-value = 0.736) immediately after the diagnosis, after six months of follow-up, and after 63 months of follow-up, respectively. Death was ILD-related in two cases (immediately after the diagnosis and after six months), whereas nine patients were lost to the follow-up immediately after the diagnosis (three anti-Jo1-positive and six non anti-Jo1-positive). The median follow-up was 36 (14–65) months. Clinical features observed during follow-up in the two groups are reported in Table 2.

During follow-up, the 47 patients underwent serial/periodic evaluation of lung function tests for a total of 180 evaluations/visits. The model-based change in FVC and DLCO in both groups of patients is reported in Figure 1, reporting no statistically significant differences among the two groups (*p*-value 0.7813 and *p*-value 0.7051, respectively), adjusting for anti-SSA/Ro52+, acute onset, diagnostic delay, and immunosuppressive treatment.

We did not observe statistically significant differences between anti-Jo1 and non-anti-Jo1 groups regarding PFTs in terms of 6-min walking distance (*p*-value = 0.8273) (Figure 2), adjusting for anti-SSA/Ro52+, acute onset, diagnostic delay, and immunosuppressive treatment.

The longitudinal multivariable model on TLC showed that anti-Jo1 patients had a TLC of 12.8% (95% confidence interval: 0.08; 25.6, *p*-value 0.0487) higher with respect to non-Jo1 patients.

There were no differences regarding the immunosuppressive treatment (Table 3) or oxygen supplementation during follow-up between the two groups (data not shown).

## 4. Discussion

We described a multicenter population of ILD patients affected by ASSD with no significant differences in age, sex, smoking habit, and respiratory functional progression between anti-Jo-1 positive and negative patients, with a median follow-up of 36 months. At diagnosis, these patients showed a mild ventilatory impairment; most of them were treated with steroids with or without immunosuppressive agents with benefit, as shown by the improvement of FVC [9,21].

In ASSD with pulmonary involvement, conflicting results are reported about different outcomes between patients with anti-Jo-1 as compared to those in which this autoantibody was absent.

Particularly, Aggarwal and colleagues showed that the survival of non-anti-Jo1 patients was worse than that of patients showing the anti-Jo-1 autoantibody [13]. Similarly, other studies observed that patients showing a positivity to PL-7 or PL-12 had a more severe ILD compared to anti-Jo1 patients [12,22]. Similarly, Hervier and colleagues reported a median FVC value that was significantly lower in both anti-PL7/PL12 compared to anti- Jo-1 positive patients [23].

On the contrary, a recent large study, including 828 ASSD patients (2), showed that survival was not influenced by the underlying antisynthetase antibody positivity. Also, Gonzalez-Perez and colleagues observed no differences between anti-synthetase patients with and without anti-Jo-1 autoantibody in 118 patients with ILD and positivity to one anti-synthetase antibody in terms of lung function during a follow up of 749 days [10]. No differences were observed in another smaller study analyzing 21 patients [24].

Our study confirms the findings observed by Gonzalez-Perez et al. for a larger population [10]. However, the majority of the studies analyzed lung function tests comparing anti-Jo1 and non-anti-Jo1 at baseline. As far as we know, our study is the first that compares the functional progression of anti-Jo1 and non-anti-Jo1 positive patients (including anti-PL12, -PL7, -EJ, and -OJ) for a period of follow-up longer than 2 years.

When analyzing TLC, we observed a borderline, although a slightly significant difference in progression between anti-Jo1 and non-anti-Jo1. The data relating to the evolution of TLC is difficult to explain due to the low number of data analyzed and should be re-evaluated in a wider study, considering a greater number of patients. However, in the functional assessment of interstitial lung diseases, FVC and DLCO represent more robust indicators of disease progression and therefore, data regarding TLC should be considered as marginal.

In our study, we observed a statistically significant increase in anti-SSA/Ro52 + autoantibodies in patients with ASS, particularly in the group detecting autoantibodies other than anti-Jo-1. The role of anti-SSA/Ro52 autoantibodies in ASS is debated, with few evidences showing that these antibodies are more frequently associated with an acute or severe presentation without influencing the patient’s prognosis [4,5]. In our study, anti-SSA/Ro52+ are more frequent in non-anti-Jo-1 patients and its presence does not influence the longitudinal multivariable mixed model of the two main outcome variables, FVC and DLCO, in the two groups of patients.

The limitations of our study are mainly related to the retrospective design and to the limited number of patients enrolled in the study. Another limitation could have been represented by the absence of definite criteria to diagnose ASS, with the possibility that different Centers could have utilized heterogeneous diagnostic criteria, possibly inducing high variability in disease duration, clinical and functional features, and duration of follow-up. Although the different role of rheumatologists and pulmonologists may have caused a different evaluation in the diagnostic process, the involvement of a limited number of tertiary Centers could have counteracted the absence of definite and univocal sets of criteria to diagnose ASSD. Nevertheless, in our study population, we did not define the various radiological patterns occurring in our patients, introducing a potential bias related to the possibility that a specific radiological pattern may be a risk factor that has driven the natural history of the disease.

In conclusion, our study confirms the absence of differences in pulmonary functional disease progression of the two subgroups of patients (anti-Jo-1 vs. non anti-Jo-1 positivity), suggesting that the type of autoantibody detected in the framework of ASS does not affect lung function decline. Therefore, the frequency of the follow-up should not be driven by specific ASS autoantibodies.

## Figures and Tables

**Figure 1 jcm-09-03033-f001:**
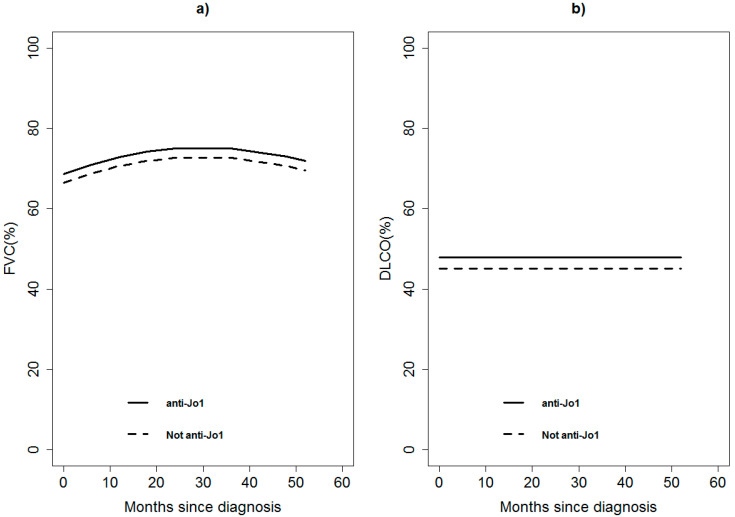
Model-based predicted value of forced vital capacity (FVC, panel **a**) and carbon monoxide diffusion capacity (DLCO, panel **b**) by time and anti-Jo1 for a patient 60 years old, anti-SSA/Ro52+ negative, acute onset, diagnosis within 12 months from symptom onset, and without immunosuppressant treatment. A continuous line represents anti-Jo1 patients; dashed lines represent non anti-Jo1 patients.

**Figure 2 jcm-09-03033-f002:**
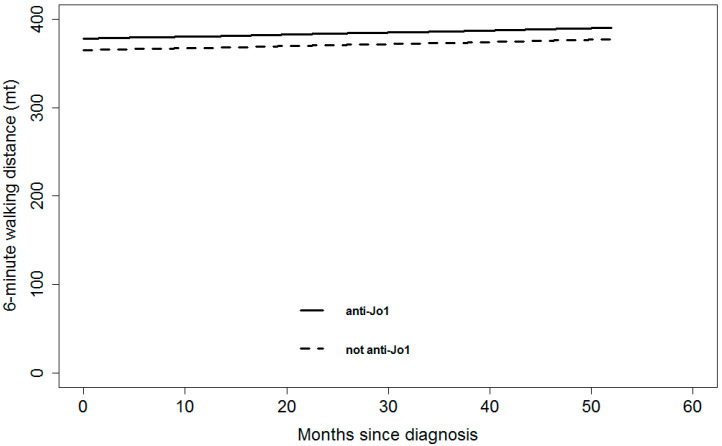
Model-based predicted value of 6-min walking distance by time and anti-Jo1 for a patient 60 years old, anti-SSA/Ro52+ negative, acute onset, diagnosis within 12 months from symptom onset, and without immunosuppressant treatment. A continuous line represents anti-Jo1 patients; dashed lines represent non anti-Jo1 patients.

**Table 1 jcm-09-03033-t001:** Demographic, clinical, serological features, and lung function parameters at baseline.

Parameters	General Population	Jo1	Non Jo1	*P*
*N*	57	32	25	
sex = F (%)	39 (68.4)	20 (62.5)	19 (76.0)	0.391
Age of development of respiratory symptoms (median (IQR))	63 (50,70)	61 (51,70)	65 (50,69)	0.843
Age of diagnosis ASS (median (IQR))	65.5 (52,71)	61 (52,71)	66 (55,71)	0.723
Delay on diagnosis in months (median (IQR))	7 (2,20)	6 (1,16)	10 (3,20)	0.095
Smoking status—active smoker or former smoker (%)	15 (27.8)	6 (20.0)	9 (37.5)	0.223
Coexistance of other autoimmune diseases (%)	16 (28.1)	7 (21.9)	9 (36.0)	0.373
**Antibodies**	**General Population**	**Jo1**	**Non Jo1**	***P***
ANA (%)	31 (60.8)	16 (57.1)	15 (65.2)	0.58
Anti-SSA/Ro 52+ (%)	26 (56.5)	10 (40.0)	16 (76.2)	0.019
**Symptoms**	**General Population**	**Jo1**	**Non Jo1**	***P***
Exertional dyspnea (%)	46 (83.6)	25 (80.6)	21 (87.5)	0.716
Dyspnea at rest (%)	17 (29.8)	10 (31.2)	7 (28.0)	1
Cough (%)	30 (53.6)	15 (46.9)	15 (62.5)	0.288
Chest pain (%)	3 (5.4)	2 (6.2)	1 (4.2)	1
Raynaud’s phenomenon (%)	9 (16.1)	2 (6.2)	7 (29.2)	0.03
Fever (%)	15 (26.8)	6 (18.8)	9 (37.5)	0.138
Mechanic’s hands (%)	3 (5.6)	2 (6.2)	1 (4.5)	1
Joint involvement (%)	23 (40.4)	13 (40.6)	10 (40.0)	1
Muscle involvement (%)	17 (29.8)	11 (34.4)	6 (24.0)	0.561
Skin lesions (%)	11 (19.3)	4 (12.5)	7 (28.0)	0.184
GI tract symptoms (%)	4 (7.0)	3 (9.4)	1 (4.0)	0.623
Systemic symptoms (%)	15 (26.3)	7 (21.9)	8 (32.0)	0.546
Chronic onset of respiratory symptoms (%)	27 (49.1)	14 (46.7)	13 (52.0)	789
Respiratory symptoms as first onset of ASS (%)	42 (75.0)	22 (71.0)	20 (80.0)	0.542
	**General Population**	**Jo1**	**Non Jo1**	***P***
TLC (%) (median (IQR))	77% (66,88)	79% (74,88)	74 (61,88)	0.265
FVC (%) (median (IQR))	80 (59,92,5)	78% (60,92)	81 (58,92)	0.941
VC (%) (median (IQR))	77% (60,92)	78% (66,92)	60 (58,88)	0.295
DLCO (%) (median (IQR))	46% (37,57)	43% (37,52)	52 (34,58)	0.7
IT (%) (median (IQR))	80.5 (77,86.5)	80% (77,84)	81 (75,86)	0.887
Walking test (m) (median (IQR))	352 (299,460)	385 (319,490)	342 (281,412)	0.334
Walking test (% minimal SaO_2_) (median (IQR))	92 (88,94)	92 (89,94)	92 (87,95)	0.815
Walking test (% median SaO_2_) (median (IQR))	92 (89,96)	92 (90,96)	92 (88,96)	0.776
	**General Population**	**Jo1**	**Non Jo1**	***P***
Corticosteroids (%)	50 (90.9)	29 (90.6)	21 (91.3)	1
Immunosuppressant	27 (50.9)	19 (59.4)	8 (38.1)	0.166
Cyclosporin A	4	3	1	
Azathioprine	8	4	4	
Methotrexate	4	3	1	
Cyclophosphamide	8	7	1	
Cyclosporin A + cyclophosphamide	2	1	1	
Rituximab	1	1	-	
Oxygen therapy (%)	15 (27.8)	9 (28.1)	6 (27.3)	1

**Table 2 jcm-09-03033-t002:** Clinical features during follow up.

Clinical Features During Follow Up	Jo1	NonJo1	*P*
*N*	29	18	
Raynaud’s phenomenon (%)	1 (3.4)	0 (0.0)	1.000
Fever (%)	1 (3.4)	0 (0.0)	1.000
Mechanic’s hands (%)	3 (10.7)	3 (16.7)	0.666
Joint pain (%)	5 (17.9)	6 (33.3)	0.296
Muscle pain (%)	3 (10.7)	3 (16.7)	0.666
Skin lesions (%)	4 (14.3)	2 (11.1)	1.000
Systemic symptoms (%)	3 (10.7)	2 (11.8)	1.000
Acute exacerbation of respiratory symptoms (%)	8 (29.6)	5 (29.4)	1.000

**Table 3 jcm-09-03033-t003:** Longitudinal multivariable mixed model of forced vital capacity (FVC) and carbon monoxide diffusion capacity (DLCO) between anti-Jo1 and non-anti-Jo1 during follow up.

	FVC (%)		DLCO (%)	
Variables	Estimate (95%Confidence Interval)	*P*-Value	Estimate (95% Confidence Interval)	*P*-Value
anti-Jo1 vs. non-anti-Jo1	2.31 (−14.13;18.74)	0.7813	2.70 (−7.46;16.86)	0.7051
Anti-SSA/Ro52+ Yes vs. No	7.05 (−8.10;22.20)	0.3579	3.49 (−10.27;17.26)	0.6153
Acute onset vs. chronic	3.64 (−10.76;18.03)	0.6072	0.63 (−12.29;13.55)	0.9230
Delay in diagnosis (>1 year vs. within 1 years)	5.58 (−11.00;22.17)	0.5058	0.66 (−14.18;15.51)	0.9296
Immunosuppressant treatment (Yes vs. No)	8.03 (−12.56;28.61)	0.4409	−2.85 (−19.14;13.43)	0.7286

The estimate represents the difference in the outcome (i.e., FVC and DLCO) among characteristics reported in the first column (variables) at any time of the follow-up other characteristics being equal. For example, the anti-Jo1 patients show a FVC of 2.31(95%CI: −14.13;18.74) percentage points higher (but not statistically significant) than non-anti-Jo1 patients with the same other characteristics (month since diagnosis, age at diagnosis, anti-SSA/Ro52+, acute onset, delay in diagnosis, and immunosuppressant treatment) at any time of the follow-up. This result is shown in Figure 1 by the trend over time of FVC for anti-Jo1 and non-anti-Jo1 patients (the two curves differ by 2.31% at any follow-up time). Figure 1 also shows the trend in time of FVC guided by the variable month (and squared month). DLCO shows a constant value in time, as visible in Figure 1, and from the estimate of month since diagnosis in the model, every month DLCO increased by 0 (95%CI: −0.11;0.11) percentage points on average.

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
