# Peer review of "Functional Progression in Patients with Interstitial Lung Disease Resulted Positive to Antisynthetase Antibodies: A Multicenter, Retrospective Analysis"

_jcm, 2020, doi:10.3390/jcm9093033_

Round 1
Reviewer 1 Report
The authors describe an excellent report on the pulmonary progression of patients with Jo-1 and non-Jo-1 antisynthetase syndrome. They suggest that the antibody subtype does not affect lung function decline. The paper is well written, is sound scientifically and will be of interest to both Respiratory and Rheumatology colleagues.
Dei and colleagues described the results of a multicentre retrospective study of the lung function decline in patients with anti-synthetase syndrome (ASSD) associated with Jo-1 and non Jo-1 antibodies. Existing literature conflicts as to whether there are phenotypic differences between patients with Jo-1 and non-Jo-1 related ASSD. Little existing literature has specifically examined the role of antibody profile in ILD progression. The results suggest that there are no phenotypic differences in the natural history of lung function progression over time in patients with Jo-1 versus anti-Jo1.
The article is well written and the tables are presented in a format that is easily understandable to the reader.
The authors could expand the discussion by highlighting the limitation of not defining the ILD pattern in their cohort
Author Response
R1.C1
The authors describe an excellent report on the pulmonary progression of patients with Jo-1 and non-Jo-1 antisynthetase syndrome. They suggest that the antibody subtype does not affect lung function decline. The paper is well written, is sound scientifically and will be of interest to both Respiratory and Rheumatology colleagues.
R1.R1
We thank the Referee for his/her appreciation of our manuscript.
R1.C2
The authors could expand the discussion by highlighting the limitation of not defining the ILD pattern in their cohort.
R1.R2
We thank the Referee for the comment. In the paragraph of the discussion regarding limitations, we highlighting the limitation of not defining the ILD pattern in our cohort.
Reviewer 2 Report
In this study, Dei et al studied the progression of interstitial lung disease in Antisynthetase syndrome patients. The study is different from other studies on the subject, as the authors monitored the progression of lung disease in patients and compared the effects of Jo1 antibody vs Non-Jo1 antibodies on lung disease.
In the study, the authors do not mention the monitoring protocols for the patients. There is no information about how often the patients had PFT’s and what was the monitoring period. They report follow up periods varied between 14 to 65 months which has a very wide range and it may impact the results due to the small sample size.
In the results, the authors mention no significant difference in the 6 min walk test, and I think it will be helpful to add those figures to the results.
Overall the study demonstrates there is no significant statistical difference in the progression of lung disease in the patients based on their underlying antisynthetase antibodies.
This study offers a good foundation for further studies to study outcomes of interstitial lung disease and explore the effects of different antibodies on the long term prognosis in antisynthetase syndrome.
Author Response
R2.C1
In the study, the authors do not mention the monitoring protocols for the patients. There is no information about how often the patients had PFT’s and what was the monitoring period. They report follow up periods varied between 14 to 65 months which has a very wide range and it may impact the results due to the small sample size.
R2.R1
This is an important point and we thank the Referee for raising it. In the revised version of the manuscript, in the “Methods” section (paragraph: study design), we added the monitoring protocol, specifically characterized by lung function test performance.
R2.C2
In the results, the authors mention no significant difference in the 6 min walk test, and I think it will be helpful to add those figures to the results.
R2.R2
We thank the Referee for the comment. We added a figure (Figure2) describing the results of the 6-minute walking test during follow-up.
R2.C3
This study offers a good foundation for further studies to study outcomes of interstitial lung disease and explore the effects of different antibodies on the long term prognosis in antisynthetase syndrome.
R2.R3
We thank the Referee for the appreciation of our manuscript.